# Bayesian multi-domain learning for cancer subtype discovery from next-generation sequencing count data

**Ehsan Hajiramezanali**
Texas A&M University
ehsanr@tamu.edu

**Siamak Zamani Dadaneh**
Texas A&M University
siamak@tamu.edu

**Alireza Karbalayghareh**
Texas A&M University
alireza.kg@tamu.edu

**Mingyuan Zhou**
University of Texas at Austin
Mingyuan.Zhou@mccombs.utexas.edu

**Xiaoning Qian**
Texas A&M University
xqian@ece.tamu.edu

## Abstract

Precision medicine aims for personalized prognosis and therapeutics by utilizing recent genome-scale high-throughput profiling techniques, including next-generation sequencing (NGS). However, translating NGS data faces several challenges. First, NGS count data are often overdispersed, requiring appropriate modeling. Second, compared to the number of involved molecules and system complexity, the number of available samples for studying complex disease, such as cancer, is often limited, especially considering disease heterogeneity. The key question is whether we may integrate available data from all different sources or domains to achieve reproducible disease prognosis based on NGS count data. In this paper, we develop a Bayesian Multi-Domain Learning (BMDL) model that derives domain-dependent latent representations of overdispersed count data based on hierarchical negative binomial factorization for accurate cancer subtyping even if the number of samples for a specific cancer type is small. Experimental results from both our simulated and NGS datasets from The Cancer Genome Atlas (TCGA) demonstrate the promising potential of BMDL for effective multi-domain learning without "negative transfer" effects often seen in existing multi-task learning and transfer learning methods.

## 1 Introduction

In this paper, we study **Bayesian Multi-Domain Learning (BMDL)** for analyzing count data from next-generation sequencing (NGS) experiments, with the goal of enhancing cancer subtyping in the *target domain* with a limited number of NGS samples by leveraging surrogate data from other domains, for example relevant data from other well-studied cancer types. Due to both biological and technical limitations, it is often difficult and costly, if not prohibitive, to collect enough samples when studying complex diseases, especially considering the complexity of disease processes. When studying one cancer type, there are typically at most hundreds of samples available with tens of thousands of genes/molecules involved, including in the case of the arguably largest cancer consortium, The Cancer Genome Atlas (TCGA) [The Cancer Genome Atlas Research Network et al., 2008]. Considering the heterogeneity in cancer and the potential cost of clinical studies and profiling, we usually have only less than one hundred samples, which often does not lead to generalizable results. Our goal here is to develop effective ways to derive predictive feature representations using available NGS data from different sources to help accurate and reproducible cancer subtyping.

The assumption of having only one domain is restrictive in many practical scenarios due to the nonstationarity of the underlying system and data heterogeneity. Multi-task learning (MTL), transfer learning (TL), and domain adaptation (DA) techniques have recently been utilized to leverage the

relevant data and knowledge of different domains to improve the predictive power in all domains or one target domain [Pan and Yang, 2010, Patel et al., 2015]. In MTL, there are $D$ different labeled domains where data are related and the goal is to improve the predictive power of all domains altogether. In TL, there are $D-1$ source domains and one target domain such that we have plenty of labeled data in the source domains and a few labeled data in the target domain, and the goal is to take advantage of source data, for example by domain adaptation, to improve the predictive power in the target domain. Although many TL and MTL methods have been proposed, "negative transfer" may happen with degraded performance when the domains are not related but the methods force to "transfer" the data and model knowledge. There still lacks a rigorous theoretical understanding when data from different domains can help each other due to the discriminative nature of these methods.

In this paper, instead of following most of TL/MTL methods relying on discriminative models $p(y|\theta, \vec{n})$ given high-dimensional count data $\vec{n}$, we propose a generative framework to learn more flexible latent representations of $\vec{n}$ from different domains. We first construct a Bayesian hierarchical model $p(\vec{n})$, which is essentially a factorization model for counts $\vec{n}$, to derive domain-dependent latent representations allowing both domain-specific and globally shared latent factors. Then the learned low-dimensional representations can be used together with any supervised or unsupervised predictive models for cancer subtyping. Due to its unsupervised nature when deriving latent representations, we term our model as Bayesian Multi-Domain Learning (BMDL). This is desirable in cancer subtyping since we may not always have labeled data and thus the model flexibility of BMDL enables effective transfer learning across different domains, with or without labeled data.

By allowing the assignment of the inferred latent factors to each domain independently based on the amount of contribution of each latent factor to that domain, BMDL can automatically learn the sample relevance across domains based on the number of shared latent factors in a data-driven manner. On the other hand, the domain-specific latent factors help keep important information in each domain without severe information loss in the derived domain-dependent latent representations of the original count data. Therefore, BMDL automatically avoids "negative transfer" with which many TL/MTL methods are dealing. At the same time, the number of shared latent factors can serve as one possible measure of domain relevance that may lead to more rigorous theoretical study of TL/MTL methods.

Specifically, for BMDL, we propose a novel multi-domain negative binomial (NB) factorization model for over-dispersed NGS count data. Similar as Dadaneh et al. [2018] and Hajiramezanali et al. [2018], we employ NB distributions for count data to obviate the need for multiple *ad-hoc* pre-processing steps as required in most of gene expression analyses. More precisely, BMDL identifies domain-specific and globally shared latent factors in different sequencing experiments as domains, corresponding to gene modules significant for subtyping different cancer types for example, and then use them to improve subtyping performance in a target domain with a very small number of samples. We introduce latent binary "selector" variables which help assign the factors to different domains. Inspired by Indian Buffet Process (IBP) [Ghahramani and Griffiths, 2006], we impose beta-Bernoulli priors over them, leading to sparse domain-dependent latent factor representations. By exploiting a novel data augmentation technique for the NB distribution [Zhou and Carin, 2015], an efficient Gibbs sampling algorithm with closed-form updates is derived for BMDL. Our experiments on both synthetic and real-world RNA-seq datasets verify the benefits of our model in improving predictive power in domains with small training sets by borrowing information from domains with rich training data. In particular, we demonstrate a substantial increase in cancer subtyping accuracy by leveraging related RNA-seq datasets, and also show that in scenarios with unrelated datasets, our method does not create adverse effects.

## 2   Related work

TL/MTL methods typically assume some notions of relevance across domains of the corresponding tasks: All tasks under study either possess a cluster structure [Xue et al., 2007, Jacob et al., 2009, Kang et al., 2011], share feature representations in common low-dimensional subspaces [Argyriou et al., 2007, Rai and Daume III, 2010], or have parameters drawn from shared prior distributions [Chelba and Acero, 2006]. Most of these methods force the corresponding assumptions for MTL to link the data across domains. However, when tasks are not related to the corresponding data from different underlying distributions, forcing MTL may lead to degraded performance. To solve this problem, Passos et al. [2012] have proposed a Bayesian nonparametric MTL model by representing the task parameters as a mixture of latent factors. However, this model requires the number of both

latent factors and mixtures to be less than the number of domains. This may lead to information loss and the model only has shown advantage when the number of domains is high. But in real-world applications, when analyzing cancer data for example, we may only have a small number of domains. Kumar and Daume III [2012] have assumed the task parameters within a group of related tasks lie in a low-dimensional subspace and allowed the tasks in different groups to overlap with each other in one or more bases. But this model requires a large number of training samples across domains.

The hierarchical Dirichlet process (HDP) [Teh et al., 2005] has been proposed to borrow statistical strengths across multiple groups by sharing mixture components. Although HDP is aimed for a general family of distributions, to make it more suitable for modeling count data, special efforts pertaining to the application need to be carried out. To directly model the counts assigned to mixture components as NB random variables, Zhou and Carin [2015] have performed a joint count and mixture modeling via the NB process. Under the NB process and integrated to HDP [Teh et al., 2005], NB-HDP employed a Dirichlet process (DP) to model the rate measure of a Poisson process. However, NB-HDP is constructed by fixing the probability parameter of NB distribution. While fixing the probability parameter of NB is a natural choice in mixture modeling, where it appears irrelevant after normalization, it would make a restrictive assumption that each count vector has the same variance-to-mean ratio. This is not proper for NGS count modeling in this paper. Closely related to the multinomial mixed-membership models, Zhou [2018] have proposed the hierarchical gamma-negative binomial process (hGNBP) to support countably infinite factors for negative binomial factor analysis (NBFA), where each of the sample $J$ is assigned with a sample-specific GNBP and a globally shared gamma process is mixed with all the $J$ gamma-negative binomial Markov chains (GMNBs). Our BMDL also uses hGNBP to model the counts in each domain, but imposes a spike and slab model to ensure domain-specific latent factors can be identified.

In this paper, we propose a hierarchical Bayesian model—BMDL—for multi-domain learning by deriving domain-dependent latent representations of observed data across domains. By jointly deriving latent representations with both domain-specific and shared latent factors, we take the best advantage of shared information across domains for effective multi-domain learning. In the context of cancer subtyping, we are interested in deriving such meaningful representations for accurate and reproducible subtyping in the target domain, where only a limited number of samples are available. We will show first in our experiments that when the source and target data share more latent factors, we can better help subtyping in the target domain with higher accuracy; more importantly, we will also show that even when the domains are distantly related, our method can selectively integrate the information from other domain(s) to improve subtyping in the target domain while prohibit using irrelevant knowledge to avoid performance degradation.

## 3 Method

We would like to model the observed counts $n_{vj}^{(d)}$ from next-generation sequencing (NGS) for gene $v \in \{1, ..., V\}$ in sample $j \in \{1, ..., J_d\}$ of domain $d \in \{1, ..., D\}$ to help cancer subtyping. The main modeling challenges here include: (1) NGS counts are often over-dispersed and requiring

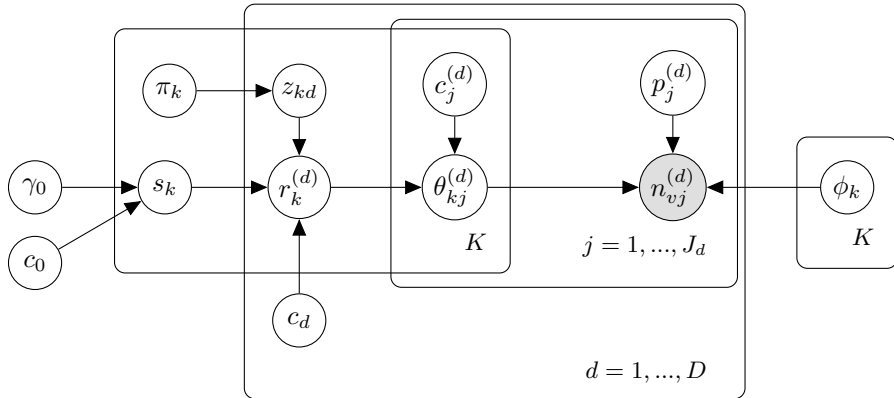

Figure 1: BMDL based on multi-domain negative binomial factorization model.

*ad-hoc* pre-processing that may lead to biased results; (2) there are a much smaller number of samples with respect to the number of genes ($V \gg J$), especially in the target domain of interest; and (3) it is often unknown how relevant/similar the samples across different domains are so that forcing the joint learning may lead to degraded performance.

We construct a Bayesian Multi-Domain Learning (BMDL) framework based on a domain-dependent latent negative binomial (NB) factor model for NGS counts so that (1) over-dispersion is appropriately modeled and *ad-hoc* pre-processing is not needed; (2) low-dimensional representations of counts in different domains can help achieve more robust subtyping results; and most importantly, (3) the sample relevance across domains can be explicitly learned to guarantee the effectiveness of joint learning across multiple domains.

BMDL achieves flexible multi-domain learning by first constructing a NB factorization model of NGS counts, and then explicitly establishing the relevance of samples across different domains by introducing domain-dependent binary variables that assign latent factors to each domain. The graphical representation of BMDL is illustrated in Fig. 1.

We model NGS counts $n_{vj}^{(d)}$ based on the following representations

$$n_{vj}^{(d)} = \sum_{k=1}^{K} n_{vjk}^{(d)}, \quad n_{vjk}^{(d)} \sim \text{NB}\left(\phi_{vk}\theta_{kj}^{(d)}, p_j^{(d)}\right), \tag{1}$$

where $n_{vj}^{(d)}$ is factorized by $K$ sub-counts $n_{vjk}^{(d)}$, each of which is a latent factor distributed according to a NB distribution. The factor loading parameter $\phi_{vk}$ quantifies the association between gene $v$ and latent factor $k$, while the score parameter $\theta_{kj}^{(d)}$ captures the popularity of factor $k$ in sample $j$ of domain $d$. It should be noted that the factor loadings are shared across all domains, and thus making their inference more robust when the number of samples is low, especially in the target domain. This does not put a restriction on the model flexibility in capturing the inter-domain variability as the score parameters determine the significance of corresponding latent factors across domains. The score parameter $\theta_{kj}^{(d)}$ is assumed to follow a gamma distribution:

$$\theta_{kj}^{(d)} \sim \text{Gamma}\left(r_k^{(d)}, 1/c_j^{(d)}\right), \tag{2}$$

with the scale parameter $c_j^{(d)}$ modeling the variability of sample $j$ of domain $d$ and the shape parameter $r_k^{(d)}$ capturing the popularity of factor $k$ in domain $d$. To further enable domain-dependent latent representations, we introduce another hierarchical layer on the shape parameter:

$$r_k^{(d)} \sim \text{Gamma}\left(s_k z_{kd}, 1/c_d\right), \tag{3}$$

where the set of binary latent variables $z_{kd}$ are considered as domain-dependent selector variables to allow different latent representations with the corresponding $r_k^{(d)}$ being present or absent across domains: When $z_{kd} = 1$, the latent factor $k$ is present for factorization of counts in domain $d$; and it is absent otherwise. In our multi-domain learning framework, as the sample relevance across domains can vary significantly, this layer provides the additional model flexibility to model the sample relevance in the given data across domains. In (3), $s_k$ is the global popularity of factor $k$ in all domains. Inspired by the beta-Bernoulli process [Thibaux and Jordan, 2007], whose marginal representation is also known as the Indian Buffet Process (IBP) [Ghahramani and Griffiths, 2006], and its use in nonparametric Bayesian sparse factor analysis [Zhou et al., 2009], we impose a beta-Bernoulli prior to the assignment variables:

$$z_{kd} \sim \text{Bernoulli}(\pi_k), \qquad \pi_k \sim \text{Beta}(c/K, c(1 - 1/K)), \tag{4}$$

which can be seen as an infinite spike-and-slab model as $K \to \infty$, where the spikes are provided by the beta-Bernoulli process and the slab is provided by the top-level gamma process. As a result, the proposed model assigns positive probability to only a subset of latent factors, selected independently of their masses.

We further complete the hierarchical Bayesian model for multi-domain learning by placing appropriate priors on the model parameters in (1), (2), (3) and (4):

$$(\phi_{1k}, \dots, \phi_{Vk}) \sim \text{Dir}(\eta, \dots, \eta), \quad \eta \sim \text{Gamma}(s_0, w_0), \quad p_j^{(d)} \sim \text{Beta}(a_0, b_0),$$

$$c_j^{(d)} \sim \text{Gamma}(e_0, 1/f_0), \quad c_d \sim \text{Gamma}(h_0, 1/u_0), \quad s_k \sim \text{Gamma}(\gamma_0/K, 1/c_0),$$

$$\gamma_0 \sim \text{Gamma}(a_0, 1/b_0), \quad c_0 \sim \text{Gamma}(s_0, 1/t_0). \tag{5}$$

From a biological perspective, $K$ factors may correspond to the underlying biological processes, cellular components, or molecular functions causing cancer subtypes, or more generally different phenotypes or treatment responses in biomedicine. The corresponding sub-counts $n_{vjk}^{(d)}$ can be viewed as the result of the contribution of underlying biological process $k$ to the expression of gene $v$ in sample $j$ of domain $d$. The probability parameter $p_j^{(d)}$, which depends on the sample index, accounts for the potential effect of varying sequencing depth of sample $j$ in domain $d$. More precisely, the expected expression of gene $v$ in sample $j$ and domain $d$ is $\sum_{k=1}^{K} \phi_{vk}\theta_{kj}^{(d)} \frac{p_j^{(d)}}{1-p_j^{(d)}}$, and hence the term $(\sum_{k=1}^{K} \phi_{vk}\theta_{kj}^{(d)})$ can be viewed as the true abundance of gene $v$ in domain $d$, after adjusting for the sequencing depth variation across samples. Specifically, it comprises of contributions from both domain-dependent and globally shared latent factors, where the amount of contribution of each latent factor can automatically be learned for the sample relevance across domains.

Given the BMDL model in Fig. 1, we derive an efficient Gibbs sampling algorithm with closed-form updating steps for inferring the model parameters by exploiting the data augmentation technique in Zhou and Carin [2015]. The detailed Gibbs sampling procedure is provided in the *supplemental materials*.

For real-world NGS datasets that are deeply sequenced and thus possess large counts, the steps in Gibbs sampling involving the Chinese Restaurant Table (CRT) distribution in Zhou and Carin [2015] are the source of main computational burden. To speed up sampling from CRT, we propose the following scheme: to draw $\ell \sim \text{CRT}(n, r)$, when $n$ is large, we first draw $\ell_1 \sim \text{CRT}(m, r)$, where $m \ll n$. Then, we draw $\ell_2 \sim \text{Pois}(r[\psi(n + r) - \psi(m + r)])$, where $\psi(\cdot)$ is the digamma function. Finally, we have $\ell \approx \ell_1 + \ell_2$. This approximation is inspired by Le Cam's theory [Le Cam, 1960], and reduces the number of Bernoulli draws required for CRT from $n$ to $m$, and hence speeding up the Gibbs sampling substantially in our experiments with TCGA NGS data, where it is not uncommon for $n > 10^5$.

## 4 Experimental Results

To verify the advantages of our BMDL model with the flexibility to capture the varying sample relevance across domains with both domain-specific and globally shared latent factors, we have designed experiments based on both simulated data and RNA-seq count data from TCGA [The Cancer Genome Atlas Research Network et al., 2008]. We have implemented BMDL to extract domain-dependent low-dimensional latent representations and then examined how well using these extracted representations in an unsupervised manner can subtype new testing samples. We also have compared the performance of BMDL to other Bayesian latent models for multi-domain learning, including

• **NB-HDP** [Zhou and Carin, 2012], for which all domains are assumed to share a set of latent factors. This is done by involving a simple Bayesian hierarchy where the base measure for the child DPs is itself distributed according to a DP. It assumes the probability parameter of NB is fixed at $p_j^{(d)} = 0.5$.

• **HDP-NBFA**: To have fair comparison and make sure that the superior performance of BMDL is not only due to the modeling of the sequencing depth variation across samples, we apply HDP to model latent scores in NB factorization as well. More specifically we model count data as $n_{jk}^{(d)} \sim \text{NB}(\phi_k\theta_{kj}^{(d)}, p_j^{(d)})$, where $\theta_{kj}^{(d)}$ is hierarchical DP instead of hierarchical gamma process in our model. Fixing $c_j^{(d)} = 1$ in ( 2) is considered here to construct an HDP, whose group-level DPs are normalized from gamma processes with the scale parameters as $1/c_j^{(d)} = 1$.

• **hGNBP** [Zhou, 2018]: To evaluate the advantages of the beta-Bernoulli modeling in BMDL, we compare the results with hGNBP, which models count data as $n_{jk}^{(d)} \sim \text{NB}(\phi_k\theta_{kj}^{(d)}, p_j^{(d)})$. Here, $\theta_{kj}^{(d)}$ is a hierarchical gamma process and the parameter $z_{kd}$ in (4) is set to 1.

We illustrate that BMDL leads to more effective target domain learning compared to both HDP and hGNBP based models by assigning domain-specific latent factors to domains (using the beta-Bernoulli process) given observed samples, while learning the latent representations globally in a similar fashion as HDP and hGNBP. In addition, we also have compared with **hGNBP-NBFA** [Zhou,

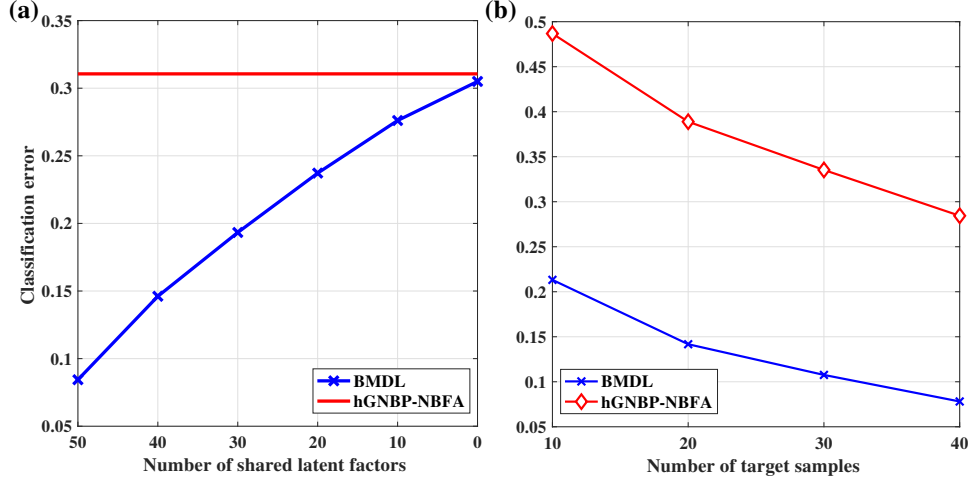

Figure 2: The classification error of BMDL and hGNBP-NBFA as a function of (a) domain relevance, and (b) the number of target samples.

2018], which can be considered as the baseline model as it extracts latent representations only using the samples from the target domain. Comparing to this baseline, we expect to show that BMDL effectively borrows the signal strength across domains to improve classification accuracy in a target domain with very small samples.

For all the experiments, we fix the truncation level $K = 100$ and consider 3,000 Gibbs sampling iterations, and retain the weights $\{r_k^{(d)}\}_{1,K}$ and the posterior means of $\{\phi_k\}_{1,K}$ as factors, and use the last Markov chain Monte Carlo (MCMC) sample for the test procedure. With these $K$ inferred factors and weights, we further apply 1,000 blocked Gibbs sampling iterations and collect the last 500 MCMC samples to estimate the posterior mean of the latent factor score $\theta_j^{(d_t)}$, for every sample of target domain $d_t$ in both the training and testing sets. We then train a linear support vector machine (SVM) classifier [Schölkopf and Smola, 2002] on all $\bar{\theta}_j^{(d_t)}$ in the training set and use it to classify each $\bar{\theta}_j^{(d_t)}$ in the test set, where $\bar{\theta}_j^{(d_t)} \in \mathbb{R}^K$ is the estimated feature vector for sample $j$ in the target domain. For each binary classification task, we report the classification accuracy based on ten independent runs. Note that although we fix $K$ with a large enough value, we expect only a small subset of the $K$ latent factors to be used and all the others to be shrunken towards zero. More precisely, inspired by the inherent shrinkage property of the gamma process, we have imposed $\mathrm{Gamma}(\gamma_0/K, 1/c_0)$ as the prior on each factor strength parameter $s_k$, leading to a truncated approximation of the gamma process using $K$ atoms.

### 4.1 Synthetic data experiments

For synthetic experiments, we compare BMDL and the baseline hGNBP-NBFA using only target samples to illustrate multi-domain learning can help better prediction in the target domain.

For the first set of synthetic data experiments, we generate the varying sample relevance across domains. The degree of relevance is controlled by varying the number of latent factors shared by the domains. In this setup, we set two domains, 1,000 features, 50 latent factors per domain, 200 samples in the source domain, and 20 samples in the target domain while the number of samples for both classes is 10. The number of shared latent factors between two domains changes from 50 to 0 to cover different degree of domain relevance. The factor loading matrix of the first domain is generated based on a Dirichlet distribution. To simulate the loading matrix for the second domain, we first select $N_{K_c}$ shared latent factors from the first domain, and then randomly generate $50 - N_{K_c}$ latent factors as unique ones for the second domain, where $N_{K_c} \in \{0, 10, 20, \ldots, 50\}$. The dispersion parameters of both domains are generated from a gamma process: $\mathrm{Gamma}(s_k, 1/c_d)$, where $s_k$ is generated by $\mathrm{Gamma}(\gamma_0/K, 1/c_0)$. The hyperparameters $\gamma_0$, and $c_0$ are drawn from $\mathrm{Gamma}(0.01, 0.01)$. To distinguish two classes of generated samples in the target domain, we generate their factor scores

by different scale parameters $c_j^{(d)} \sim \text{Gamma}(a, 0.01)$, where $a$ is set to be 100 and 150 in the first and second class, respectively. From Figure 2(a), the first interesting observation is that BMDL automatically avoids "negative transfer": the classification errors of BMDL by jointly learning the latent representations are consistently lower than the classification errors using only the target domain data no matter how many shared latent factors exist across simulated domains. Furthermore, the classification error in the target domain decreases monotonically with the number of shared latent factors, which agrees with our intuition that BMDL can achieve higher predictive power when data across domains are more relevant. This demonstrates that the number of shared latent factors across domains may serve as a new measure of the domain relevance.

In the second simulation study, we investigate how the number of target samples affects the classification performance. In this simulation setup, we simulate two related domains with 40 shared latent factors out of 50 total ones. The number of samples in the target domain is changing from 10 to 40, keeping the other setups the same as in the first experiment. Figure 2(b) shows that increasing the number of target samples will improve the performance of both the baseline hGNBP-NBFA using only target data and BMDL integrating data across domains, which is again expected. More interestingly, the improvement of BMDL over hGNBP-NBFA decreases with the number of target samples, which agrees with the general trend shown in the TL/MTL literature [Pardoe and Stone, 2010, Karbalayghareh et al., 2018] that the prediction performance eventually converges to the optimal Bayes error when there are enough samples in the target domain.

## 4.2 Case study: Lung cancer

We consider two setups of analyzing RNA-seq count data from the studies on two subtypes of lung cancer, i.e. Lung Adenocarcinoma (LUAD) and Lung Squamous Cell Carcinoma (LUSC) from TCGA [The Cancer Genome Atlas Research Network et al., 2008]. First, we take two types of NGS data, RNA-seqV2 and RNA-seq of the same lung cancer study, as two **highly-related** domains since the source and target domain difference is simply due to profiling techniques. Second, we use RNA-seq data from a Head and Neck Squamous Cell Carcinoma (HNSC) cancer study as the source domain and the above RNA-seq lung cancer data as the target domain. These are considered as **low-related** domains as these two cancer types have quite different disease mechanisms. In this set of experiments, we take 10 samples for each subtype of lung cancer in the target domain to test cancer subtyping performance. We also investigate the effect of the number of source samples, $N_s$, on cancer subtyping in the target domain by setting $N_s = 25$ and 100.

For all the TCGA NGS datasets, we first have selected the genes appeared in all the datasets and then filtered out the genes whose total read counts across samples are less than 50, resulting in roughly 14,000 genes in each dataset. We first have divided the lung cancer datasets into training and test sets, and then the differential gene expression analysis has been performed on the training set using DeSeq2 [Love et al., 2014], by which 1,000 out of the top 5,000 genes with higher log2 fold change between LUAD and LUSC have been selected for consequent analyses. We first check the subtyping accuracy by directly applying linear SVM to the raw counts in the target domain, which gives an average accuracy of 59.28% with a sample standard deviation (STD) of 5.54% from ten independent runs. We also transform the count data to standard normal data after removing the sequencing depth effect using DESeq2 [Love et al., 2014] and then apply regularized logistic regression provided by the LIBLINEAR (`https://www.csie.ntu.edu.tw/~cjlin/liblinear/`) package [Fan et al., 2008]. The classification accuracy becomes $74.10\% \pm 4.41\%$.

Table 1 provides cancer subtyping performance comparison between BMDL, NB-HDP, HDP-NBFA, hGNBP, as well as the baseline hGNBP-NBFA using only the samples form the target domain. In fact, when analyzing data across highly-related domains of lung cancer, from the identified 100 latent factors in the target domain by BMDL, there are 98 shared ones between two RNA-seq techniques. While for low-related domains of lung cancer and HMSC, only 25 of 62 extracted latent factors in lung cancer by BMDL are shared with HNSC. This is consistent with our biological knowledge regarding the sample relevance in two setups. From the table, BMDL consistently achieves better cancer subtyping in both highly- and low-related setups. On the contrary, as the results show, not only HDP based methods cannot improve the results in the low-related setup, but also the performance will be degraded with more severe "negative transfer" adversarial effects when using more source samples. The reason for this is that HDP assumes a latent factor with higher weight in the shared DP will occur more frequently within each sample [Williamson et al., 2010]. This might be an

Table 1: Lung cancer subtyping results (average accuracy (%) and STD)

| Method | highly-related ($N_s$) | | low-related ($N_s$) | |
|---|---|---|---|---|
| | 25 | 100 | 25 | 100 |
| NB-HDP | $55.22 \pm 3.69$ | $56.52 \pm 4.61$ | $54.57 \pm 7.73$ | $53.83 \pm 7.79$ |
| HDP-NBFA | $63.48 \pm 1.23$ | $65.65 \pm 4.22$ | $54.89 \pm 7.38$ | $51.83 \pm 8.32$ |
| hGNBP | $74.13 \pm 7.07$ | $77.61 \pm 3.54$ | $72.94 \pm 1.70$ | $74.55 \pm 8.84$ |
| BMDL | $\mathbf{78.46} \pm 5.97$ | $\mathbf{81.49} \pm 5.12$ | $\mathbf{78.85} \pm 4.55$ | $\mathbf{78.10} \pm 5.65$ |
| hGNBP-NBFA | $73.38 \pm 7.29$ | | | |

undesirable assumption, especially when the domains are distantly related. For example, a latent factor might not be present throughout the HNSC samples but dominant within the samples of lung cancer. HDP based methods are not able to discover these latent factors given observed samples due to the limited number of lung cancer samples. In addition to this undesirable assumption, NB-HDP does not account for the sequencing-depth heterogeneity of different samples, which may lead to biased results deteriorating subtyping performance as shown in Table 1.

HDP-NBFA explores the advantages of modeling the NB dispersion and improves over the NB-HDP due to the flexibility of learning $p_j^{(d)}$, especially in the highly-related setup. This demonstrates the benefits of inferring the sequencing depth in RNA-seq count applications. Although in highly-related setup the HDP-NBFA performance has been improved with the increasing number of source samples, we still observe the same "negative transfer" effect in the low-related setup. Again, integrating more source samples is beneficial when the samples across domains are highly relevant but it can be detrimental when the relevance assumption does not hold as both NB-HDP and HDP-NBFA force a similar structure of latent factors across domains.

The better performance of the gamma process based models compared to HDP based models, in both scenarios with low and high domain relevance, may be explained by the negative correlation structure that the Dirichlet process imposes on the weights of latent factors, while the gamma process models these weights independently, and hence allowing more flexibility for adjustment of latent representations across domains. On the other hand, when comparing the performance of BMDL and hGNBP, domain-specific latent factor assignment using the beta-Bernoulli process can be considered as the main reason for the superior performance of BMDL, especially in the low-related setup.

Compared to the baseline hGNBP-NBFA, BMDL can clearly improve cancer subtyping performance. Even using a small portion of the related source domain samples, the subtyping accuracy can be improved up around 5%. With more highly-related source samples, the improvement can be up to 8%. Compared to the HDP based methods, BMDL can achieve up to 16% improvement in the highly-related setup due to the benefits brought by the gamma process modeling of count data instead of using DP in HDP models, which forces negative correlation and restricts the distribution over latent factor abundance [Williamson et al., 2010]. Compared to hGNBP, BMDL can achieve up to 4% and 6% accuracy improvement, respectively, in highly- and low-related setups due to domain-specific latent factor assignment using the beta-Bernoulli process. Since the selector variables $z_{kd}$ in BMDL help to assign only a finite number of latent factors for each domain, it is sufficient merely to ensure that the sum of any finite subset of top-level atoms is finite. This eliminates the restrictions on factor score parameters imposed by DP, and improves subtyping accuracy since the latent factor abundance is independent.

BMDL also does not have any restriction on the number of domains and can be applied to more than two domains. To show this, we also have done another case study with three domains using both the highly- and low-related TCGA datasets. The accuracy of BMDL is $79.71\% \pm 5.32\%$ and $81.96\% \pm 4.96\%$ when using $N_s^{(d_{s1})} = N_s^{(d_{s2})} = 25$ and 100 samples for two source domains as described earlier, respectively. Compared to one source and one target domain with 25 source samples, the accuracy of using three domains has improved by 1%. Having two source domains with more samples ($N_s^{(d_{s1})} + N_s^{(d_{s2})} = 50$) leads to more robust estimation of $\phi_{vk}$ and improves the subtyping accuracy. When there are enough number of samples ($N_s^{(d_{s1})} = 100$) in highly-related domain, adding another low-related domain does not improve the subtyping results. But it is notable

that the accuracy has increased around 4% when adding the highly-related domain with 100 samples to 100 low-related samples. The results show that 1) using more domains with more samples does help subtyping in the target domain; 2) BMDL avoids negative transfer even when adding samples from low-related domains.

We would like to emphasize again that, unlike existing methods, BMDL infers the domain relevance given in the data and derive domain-adaptive latent factors to improve predictive power in the target domain, regardless of the degree of domain relevance. This is important in real-world setups when the samples across domains are distantly related or the sample relevance is uncertain. As the results have demonstrated, BMDL achieves the similar performance improvement in the low-related setup as in the highly-related setup without "negative transfer" symptom, often witnessed in existing TL/MTL methods. It shows the great potential for effective data integration and joint learning even in the low-related setup: the performance is better than competing methods as well as the baseline hGNBP-NBFA using only target samples and increasing the number of source samples does not hurt the performance.

## 5    Conclusions

We have developed a multi-domain NB latent factorization model, tailored for Bayesian multi-domain learning of NGS count data—BMDL. By introducing this hierarchical Bayesian model with selector variables to flexibly assign both domain-specific and globally shared latent factors to different domains, the derived latent representations of NGS data preserves predictive information in corresponding domains so that accurate cancer subtyping is possible even with a limited number of samples. As BMDL learns domain relevance based on given samples across domains and enables the flexibly of sharing useful information through common latent factors (if any), BMDL performs consistently better than single-domain learning regardless of the domain relevance level. Our experiments have shown the promising potential of BMDL in accurate and reproducible cancer subtyping with "small" data through effective multi-domain learning by taking advantage of available data from different sources.

**Acknowledgements** We would like to thank Dr. Sahar Yarian for insightful discussions. We also thank Texas A&M High Performance Research Computing and Texas Advanced Computing Center for providing computational resources to perform experiments in this work. This work was supported in part by the NSF Awards CCF-1553281, IIS-1812641, and IIS-1812699.

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
