[Supplementary Material · supplementary.pdf]

# Bayesian multi-domain learning for cancer subtype discovery from next-generation sequencing count data

**Ehsan Hajiramezanali**
Texas A&M University
ehsanr@tamu.edu

**Siamak Zamani Dadaneh**
Texas A&M University
siamak@tamu.edu

**Alireza Karbalayghareh**
Texas A&M University
alireza.kg@tamu.edu

**Mingyuan Zhou**
University of Texas at Austin
Mingyuan.Zhou@mccombs.utexas.edu

**Xiaoning Qian**
Texas A&M University
xqian@ece.tamu.edu

## A  Gibbs sampling inference for BMDL

We provide the detailed Gibbs sampling procedure by exploiting the augmentation techniques for negative binomial (NB) factor analysis in Zhou and Carin [2015].

**Sampling $\phi_{vk}$ and $\theta_{kj}^{(d)}$:** The NB random variable $n \sim \text{NB}(r, p)$ can be generated from a compound Poisson distribution:

$$n = \sum_{t=1}^{\ell} u_t, \ \ u_t \sim \text{Log}(p), \ \ \ell \sim \text{Pois}(-r \ln(1-p)),$$

where $u \sim \text{Log}(p)$ corresponds to the logarithmic random variable [Johnson et al., 2005], with the probability mass function (pmf) $f_U(u) = -\frac{p^u}{u \ln(1-p)}$, $u = 1, 2, ...$. As shown in Zhou and Carin [2015], given $n$ and $r$, the distribution of $\ell$ is a Chinese Restaurant Table (CRT) distribution: $(\ell | n, r) \sim \text{CRT}(n, r)$, a random variable from which can be generated as $\ell = \sum_{t=1}^{n} b_t$, with $b_t \sim \text{Bernoulli}(\frac{r}{r+t-1})$.

Utilizing the above data augmentation technique, for each observed count $n_{vj}^{(d)}$, a latent count is sampled as

$$(\ell_{vj}^{(d)} | -) \sim \text{CRT}\left(n_{vj}^{(d)}, \sum_{k=1}^{K} \phi_{vk} \theta_{kj}^{(d)}\right). \tag{1}$$

These counts can further split into latent sub-counts [Zhou, 2018] using a multinomial distribution:

$$(\ell_{vj1}^{(d)}, \ldots, \ell_{vjK}^{(d)} | -) \sim \text{Mult}\left(\ell_{vj}^{(d)}; \frac{\phi_{v1} \theta_{1j}^{(d)}}{\sum_{k=1}^{K} \phi_{vk} \theta_{kj}^{(d)}}, \ldots, \frac{\phi_{vK} \theta_{Kj}^{(d)}}{\sum_{k=1}^{K} \phi_{vk} \theta_{kj}^{(d)}}\right). \tag{2}$$

These latent counts can be generated as $\ell_{vjk}^{(d)} \sim \text{Pois}(q_j^{(d)} \phi_{vk} \theta_{kj}^{(d)})$, where $q_j^{(d)} := -\ln(1-p_j^{(d)})$. Hence, using the gamma-Poisson conjugacy, and denoting $\ell_{v.k}^{(\cdot)} = \sum_{d=1}^{D} \sum_{j=1}^{J} \ell_{vjk}^{(d)}$ and $\ell_{.jk}^{(d)} = \sum_{v=1}^{V} \ell_{vjk}^{(d)}$, $\phi_{vk}$ and $\theta_{kj}^{(d)}$ are updated as

$$(\phi_{1k}, \ldots, \phi_{Vk} | -) \sim \text{Dir}(\eta + \ell_{1.k}^{(\cdot)}, \ldots, \eta + \ell_{V.k}^{(\cdot)}); \ \theta_{kj}^{(d)} \sim \text{Gamma}\left(r_k^{(d)} + \ell_{.jk}^{(d)}, \frac{1}{c_j^{(d)} - q_j^{(d)}}\right). \tag{3}$$

**Approximation:** Rather than sampling $\ell_{vj}^{(d)}$ using (1), we can approximate it as follows to further speed up the inference procedure:

$$
\begin{aligned}
\mathrm{CRT}(n,r) &= \sum_{i=1}^{n} \mathrm{Bernoulli}\left(\frac{r}{i-1+r}\right) \\
&= \sum_{i=1}^{m} \mathrm{Bernoulli}\left(\frac{r}{i-1+r}\right) + \sum_{i=m+1}^{n} \mathrm{Bernoulli}\left(\frac{r}{i-1+r}\right) \\
&= \mathrm{CRT}(m,r) + \mathrm{Pois}(\lambda), \\
\lambda &= \sum_{i=1}^{n} \frac{r}{i-1+r} = r[\psi(n+r) - \psi(m+r)].
\end{aligned}
\tag{4}
$$

This approximation reduces the computational complexity for sampling all $\ell_{vjk}^{(d)}$ from $O[\sum_d \sum_v \sum_j n_{vj}^{(d)} K]$ to $O[\sum_d \sum_v \sum_j \min(n_{vj}^{(d)}, m)K]$, which can lead to significant computation saving for a large number of genes where large counts $n_{vj}^{(d)}$ are abundant.

**Sampling $r_k^{(d)}$, $s_k$, and $\gamma_0$:** Let $\tilde{p}_j^{(d)} = -q_j^{(d)}/(c_j^{(d)} - q_j^{(d)})$. Starting with $\ell_{\cdot jk}^{(d)} \sim \mathrm{Pois}(-q_j^{(d)}\theta_{kj}^{(d)})$, marginalizing out $\theta_{kj}^{(d)}$ leads to

$$
\ell_{\cdot jk}^{(d)} \sim \mathrm{NB}(r_k^{(d)}, \tilde{p}_j^{(d)}).
\tag{5}
$$

Based on the CRT augmentation technique:

$$
(\tilde{\ell}_{jk}^{(d)}|-) \sim \mathrm{CRT}(\ell_{\cdot jk}^{(d)}, r_k^{(d)}),
\tag{6}
$$

the Gibbs sampling update for $r_k^{(d)}$ can be written as

$$
(r_k^{(d)}|-) \sim \mathrm{Gamma}\left(z_{kd}s_k + \tilde{\ell}_{\cdot k}^{(d)}, \frac{1}{c_k - \sum_j \ln(1-\tilde{p}_j^{(d)})}\right).
\tag{7}
$$

Following a similar procedure for $s_k$, first we draw

$$
(\tilde{\tilde{\ell}}_{k}^{(d)}|-) \sim \mathrm{CRT}(\tilde{\ell}_{\cdot k}^{(d)}, s_k),
\tag{8}
$$

and then we update the conditional posterior of $s_k$ as

$$
(s_k|-) \sim \mathrm{Gamma}\left(\gamma_0/K + \sum_d \tilde{\tilde{\ell}}_{k}^{(d)}, \frac{1}{c_0 - \tilde{q}_k}\right),
\tag{9}
$$

where $\tilde{q}_k := \sum_d z_{kd} \sum_j \ln(1-\tilde{p}_j^{(d)})$. Similarly, we can update posterior of $\gamma_0$ as

$$
(\acute{\ell}_k|-) \sim \mathrm{CRT}(\sum_d \tilde{\tilde{\ell}}_{k}^{(d)}, \gamma_0/K), \quad (\gamma_0|-) \sim \mathrm{Gamma}\left(a_0 + \acute{\ell}_\cdot, \frac{1}{b_0 - \sum_k \ln(1-\tilde{q}_k)/K}\right).
\tag{10}
$$

**Sampling $z_{kd}$:** Denote $\tilde{\tilde{q}}_k^{(d)} := -\sum_j \ln(1-\tilde{p}_j^{(d)})/(c_k - \sum_j \ln(1-\tilde{p}_j^{(d)}))$. Starting with $\tilde{\ell}_{\cdot k}^{(d)} \sim \mathrm{Pois}(-z_{kd}s_k \sum_j \ln(1-\tilde{p}_j^{(d)}))$, marginalizing out $s_k$ leads to $\tilde{\ell}_{\cdot k}^{(d)} \sim \mathrm{NB}(z_{kd}s_k, \tilde{\tilde{q}}_k^{(d)})$. We can write

$$
\begin{aligned}
Pr(z_{kd}|\tilde{\ell}_{\cdot k}^{(d)} = 0) &\propto Pr(\tilde{\ell}_{\cdot k}^{(d)} = 0|z_{kd})Pr(z_{kd}) \propto (\tilde{\tilde{q}}_k^{(d)})^{z_{kd}s_k} \pi_k^{z_{kd}}(1-\pi_k)^{1-z_{kd}} \\
&\propto ((\tilde{\tilde{q}}_k^{(d)})^{s_k}\pi_k)^{z_{kd}}(1-\pi_k)^{1-z_{kd}},
\end{aligned}
\tag{11}
$$

and thus we have $Pr(z_{kd}|\tilde{\ell}_{\cdot k}^{(d)} = 0) \sim \mathrm{Bernoulli}\left(\frac{(\tilde{\tilde{q}}_k^{(d)})^{s_k}\pi_k}{(\tilde{\tilde{q}}_k^{(d)})^{s_k}\pi_k + (1-\pi_k)}\right)$. Therefore, we can update $z_{kd}$ as

$$
(z_{kd}|-) \sim \delta(\tilde{\ell}_{\cdot k}^{(d)} = 0)\mathrm{Bernoulli}\left(\frac{(\tilde{\tilde{q}}_k^{(d)})^{s_k}\pi_k}{(\tilde{\tilde{q}}_k^{(d)})^{s_k}\pi_k + (1-\pi_k)}\right) + \delta(\tilde{\ell}_{\cdot k}^{(d)} > 0).
\tag{12}
$$

**Sampling $\eta$:** . To derive the update steps for Dirichlet hyperparameters, the likelihood for $\phi_k$ is

$$\mathcal{L}(\phi_k) \propto \prod_k \text{Mult}(\ell_{1\cdot k}^{(\cdot)}, \ldots, \ell_{V\cdot k}^{(\cdot)}; \ell_{\cdot\cdot k}^{(\cdot)}, \phi_k). \tag{13}$$

Marginalizing out $\phi_k$ from (13), the likelihood for $\eta$ can be expressed as

$$\mathcal{L}(\eta) \propto \prod_k \text{DirMult}(\ell_{1\cdot k}^{(\cdot)}, \ldots, \ell_{V\cdot k}^{(\cdot)}; \ell_{\cdot\cdot k}^{(\cdot)}, \eta, \ldots, \eta). \tag{14}$$

where DirMult denotes the Dirichlet-Multinomial distribution [Zhou, 2018]. The product of $\mathcal{L}(\eta)$ and $\prod_k \text{Beta}(q_k; \ell_{\cdot\cdot k}^{(\cdot)}, \eta V)$ can be expressed as

$$\mathcal{L}(\eta)\text{Beta}(q_k; \ell_{\cdot\cdot k}^{(\cdot)}, \eta V) \propto \prod_k \prod_v \text{NB}(\ell_{v\cdot k}^{(\cdot)}; \eta, q_k), \tag{15}$$

we can further apply the data augmentation technique for the NB distribution of Zhou and Carin [2015] to derive the closed-form updates for $\eta$ as

$$(q_k|-) \sim \text{Beta}(\ell_{\cdot\cdot k}^{(\cdot)}, \eta V), \quad u_{vk} \sim \text{CRT}(\ell_{v\cdot k}^{(\cdot)}, \eta),$$

$$(\eta|-) \sim \text{Gamma}\left(s_0 + \sum_{k,v} u_{kv}, \frac{1}{w_0 - V \sum_k \ln(1 - q_k)}\right) \tag{16}$$

**Sampling $p_j^{(d)}$:** Using appropriate conditional conjugacy, we can sample the remaining parameters:

$$(p_j^{(d)}|-) \sim \text{Beta}(a_0 + \sum_v n_{vj}^{(d)}, b_0 + \sum_k \theta_{jk}^{(d)}). \tag{17}$$