[Reviews · NeurIPS 2018]

Reviewer 1



In this paper the author propose a new hierarchical Bayesian model allowing to borrow information from related domains (e.g. NGS data from patients of related cancer types) to subtype more accurately a domain of interest (e.g. cluster NGS data from patients from a specific cancer). The model assumes that the counts from a gene are drawned from a negative binomial distribution which is decomposed into subcounts (also drawned from NB distributions) resulting from different factors. Those factors are shared across different domains, and their importance (including their possible absence in some domains) is modeled through a spike and slab like prior. Once those factor scores have been learned (through a Gibbs sampling algorithm), classification algorithms can be run to subtype the target domain. The authors illustrate and compare their method both on synthetic and real-data experiments. The question of appropriately subtyping cancer is certainly very timely and from their experiments section the authors have convinced me that their approach leads to substantially better results than state of the art models. However, I think the authors could improve the clarity of the reading by giving more examples throughout the introduction and stating more clearly the global workflow of their approach: what are the parameters that really matter? How is the subtyping performed? (though this is written in the introduction it took me quite some time to understand that the information borrowed from other domains is used for a better inference on the parameters which are then used to classify the patients regardless of the model). I am not familiar enough with the domain to really assess the novelty of the contribution. From my reading this paper is a "model - prior - Gibbs sampler" paper which seems to improve the classification scores but does not provide breakthrough to the learning community. The novelty essentially seems to come from the choice of the prior which allows but does not require factors to be shared across domains. Moreover, the authors state that details on their Gibbs sampler are provided in the supplementary materials but I can only trust them as there seems to have been some mistake on uploading the supplementary materials (the actual manuscript was submitted instead). I would have liked to have some intuition on how to chose the parameter K, and how does its value affect the results, both in terms of subtyping and in terms of complexity. In general, how does the approach scale with competitors in terms of run-time? In the case study section, how many subtypes of lung cancer were considered? Have the authors tried their approach with more than two domains? (for instance it would have been nice to see the case-study with both the high and low-related datasets). There are a few typos throughout the manuscript. Here are a few that I picked: l25: including in the case of the arguably l91 This may lead to information loss l244 it should be figure 2.a) -- I thank the authors for providing an answer to my questions and I am satisfied by their response.

Reviewer 2



In this manuscript the authors introduce Bayesian Multi-Domain Learning (BMDL), a Bayesian model for learning latent spaces across related domains with high predictive power. The model’s sparsity structure allows some of the latent dimensions to be domain specific and others to be shared across domains and the authors derive an efficient blocked-gibbs sampler that allows for a negative binomial likelihood for NGS count data by using a data augmentation strategy. The authors apply BMDL to both synthetic data and real-world data, where they demonstrate their method on related domains (different RNA seq versions as an example of closely related domains, and different cancer types as an example of unrelated domains). Overall, I think the model the authors propose is elegant and the problem they address (integrating multiple ‘omics datasets to improve predictive power) is important in a field with traditionally low predictive accuracy. However, I think the authors need to further convince the reader that their method is necessary over basic supervised machine learning techniques. The baseline model of a linear SVM on raw counts is inadequate: an implementation that appropriately normalizes (e.g. TPM) and scales (N(0,1)) the counts across a variety of techniques such as regularized logistic regression or even neural nets (see e.g. http://beamandrew.github.io/deeplearning/2017/06/04/deep_learning_works.html) could be used to get a true baseline. It was difficult to fully evaluate this paper as the supplementary material submitted was identical to the main paper (I assume by mistake) - I would ask the authors submit the actual supplement as part of the rebuttal. Further comments below - The authors state “Traditional machine learning methods do not directly apply here as their fundamental assumption is that samples (training and testing) are from an identical probability distribution” - this is not addressed in the paper - the authors “calculate a latent factor score for every sample in both the training and testing sets” - these come from same distribution for both training and test set so I do not see how the assertion in the introduction is addressed? (nor indeed how you could do any inference without making similarity assumptions about the training and test set) In paragraph starting line 46 it would help if in the context of the problem the authors give examples of the domains they are targeting (RNA-seq technology version and cancer types) Line 73 the authors say they exploit a “novel” data augmentation strategy for negative binomial distributions - is this the referenced [Zhou and Carin] (in which case it is not novel) or referencing the approximation scheme on line 183 for drawing from CRTs for large N ? On line 127 it would help if the authors clarified what they mean by “cancer subtyping” - ultimately they use it to predict different cancer types based on gene expression, but it can mean other things in this context (e.g. identifying clusters of patients with high/low survival, etc). In equation 2 the Gamma distribution has as its shape the product of s with a binary latent variable z - however, this can obviously lead to a Gamma with shape = 0 for which the density is undefined. It seems like the correct form for \theta would be a mixture of a point mass at 0 and a gamma distribution Gamma(s, 1/c) where the mixture component is controlled by z? Line 223 d_t should be defined as the target domain when it is introduced Line 277 - “1000 out of the top 5000 genes with higher log2 fc between LUAD and LUSC have been selected for analysis.” How were the 1000 out of the top 5000 selected? Importantly, this invalidates the accuracy claims in table 1 - the entire dataset has been used for feature selection rather than appropriately splitting the data into training and test (& validation) sets. While this has been uniformly applied across all algorithms and it appears BMDL has higher accuracy, if the point is to predict cancer subtypes based on gene expression then this is not proven. Line 280 - what are the random SVM runs? Line 286 - “From the table, BMDL clearly outperforms all the methods...more critically, BMDL consistently achieves better cancer subtyping.” What is the difference between these two? Isn’t the performance the subtyping?

Reviewer 3



The paper deals with a very important problem, of having to work with samples that have a small space. The authors formulate a Bayesian Multi-domain learning framework where domain-specific and shared latent representations from multiple domains are learnt to enable learning a target domain with a small number of samples. The application area is cancer subtyping. The paper is well written and the modeling assumptions are sound. Comments: >>>A concern I have is how the model scales - 1) since inference is based on MCMC. How long did the simulated experiments take to run. 2) The simulated and real-world experiments only had 2 domains. Have the authors explored with more than 2 domains. Does the model scale in this setting. 3) How did you handle 14K genes in your MCMC framework - as in did you have to infer variables that had dimensions equal to the number of genes? >>>In the plate figure, you could color code the blocks that are 'your contribution', given it heavily relies upon Zhou et al (2018). It will also be helpful to place the distributions in the Figure. \alpha_0 and \beta_0 are not given on the plate model. UPDATE after rebuttal: Thank you for the response and clarifications. It is a nice neat work. You should consider placing your explanations in a supplementary, if accepted/allowed.